# RNA Epigenetics in Chronic Lung Diseases

**DOI:** 10.3390/genes13122381

**Published:** 2022-12-16

**Authors:** Xiaorui Wang, Zhihou Guo, Furong Yan

**Affiliations:** 1Department of Ophthalmology, The Second Affiliated Hospital of Fujian Medical University, Quanzhou 362002, China; 2Center for Molecular Diagnosis and Therapy, The Second Affiliated Hospital of Fujian Medical University, Quanzhou 362002, China

**Keywords:** chronic lung disease, RNA epigenetics, lung cancer, RNA methylation, treatment

## Abstract

Chronic lung diseases are highly prevalent worldwide and cause significant mortality. Lung cancer is the end stage of many chronic lung diseases. RNA epigenetics can dynamically modulate gene expression and decide cell fate. Recently, studies have confirmed that RNA epigenetics plays a crucial role in the developing of chronic lung diseases. Further exploration of the underlying mechanisms of RNA epigenetics in chronic lung diseases, including lung cancer, may lead to a better understanding of the diseases and promote the development of new biomarkers and therapeutic strategies. This article reviews basic information on RNA modifications, including *N*^6^ methylation of adenosine (m^6^A), *N*^1^ methylation of adenosine (m^1^A), *N*^7^-methylguanosine (m^7^G), 5-methylcytosine (m^5^C), 2′O-methylation (2′-O-Me or Nm), pseudouridine (5-ribosyl uracil or Ψ), and adenosine to inosine RNA editing (A-to-I editing). We then show how they relate to different types of lung disease. This paper hopes to summarize the mechanisms of RNA modification in chronic lung disease and finds a new way to develop early diagnosis and treatment of chronic lung disease.

## 1. Introduction

Chronic lung diseases include chronic obstructive pulmonary disease (COPD), pneumonia, idiopathic pulmonary fibrosis(IPF), asthma, and lung cancer [1]. Despite many new treatments for chronic lung disease, the death rate for the disease has remained nearly unchanged over the years [2]. With the update of research techniques, our understanding of the changes in the genome and signaling pathways associated with chronic lung diseases has dramatically improved. These advances allow clinicians to treat patients precisely to improve outcomes. Unfortunately, due to the vast heterogeneity in most chronic lung diseases, new tried-and-true treatments are still needed for the diagnosis and treatment of chronic lung diseases [3,4,5,6]. Epigenetics and epigenetic-based targeted therapies have begun to be applied in the clinic and have made remarkable progress.

Epigenetics is a study of molecular biology that deals with the heritable variation in gene function above the primary DNA sequence. It regulates gene expression without changing the DNA sequence. The properties of cells and the differences between different types of cells often rely on systems without DNA variation. Epigenetics includes but is not limited to four principal mechanisms: DNA and RNA methylation, chromatin remodeling, noncoding RNAs, and histone modifications [7]. The development of approaches for the detection of RNA modifications has played a crucial role in the field of RNA modification research. Methylated RNA immunoprecipitation sequencing (MeRIP-Seq), as a detection method of m6A modification, can detect the presence of m6A in the whole genome [8]. Sequencing technology based on bisulfite has been widely used for the identification of m5C [9]. Pseudouridine sequencing (Pseudo-seq) is a method for pseudouracil recognition in genome-wide single nucleotide resolution. This method can accurately locate the one-base resolution at the full transcriptome level [10]. The m1A detection methods use antibodies that can specifically recognize m1A to enrich the RNA containing m^1^A, and then detects m^1^A on the mRNA [11,12]. As the approaches for the detection of RNA modifications continue to improve, more and more RNA modifications are being discovered.

RNA epigenetics has introduced a new layer of gene regulation in the study of chronic lung diseases. It dynamically regulates gene expression through a series of different modifications, broadening the potential of epigenetics in the diagnosis and treatment of chronic lung diseases [13]. Currently, a number of articles have summarized the relationship between RNA modification and respiratory diseases [14,15,16]. The majority of them have focused on lung cancer and m^6^A. With the progress of the technologies, more and more studies have confirmed the relationship between lung diseases other than lung cancer and other types of RNA modification. The computational work has made a great contribution to the development of RNA modification research and provided a valuable data set for chronic lung diseases related analysis. This review therefore also covers a number of known database/functional tools to make this review more informative and useful to the readers. This review aims to explore the mechanisms of epigenetic modifications associated with RNA, especially the impact of these modifications on chronic lung diseases.

## 2. RNA Epigenetics Mechanisms

RNA epigenetics research has developed rapidly in the last decade. RNAs participate in many processes, such as transcription, splicing, and translation. RNA regulates gene expression not only through the form of an intermediate in protein synthesis (messenger RNA (mRNA)) or an effector molecule (transfer RNA (tRNA) and ribosomal RNA (rRNA)), but also acts directly on gene expression, including through the action of multiple classes of other noncoding RNAs(ncRNAs), such as microRNA (miRNA), small nuclear RNA (snRNA), small nucleolar RNA (snoRNA) and long ncRNA (lncRNA) [17,18,19]. RNA is a novel function as a catalyst and regulator of many biochemical reactions, a carrier of genetic information, an adaptor for protein synthesis, and a structural scaffold for subcellular organelles [20,21,22,23,24]. RNA epigenetics is generally considered to be irreversible changes that have significant effects on RNA structure stability and/or function. However, some RNA modifications are reversible [25,26]. With only 4 bases, RNA is less diverse than protein with 20 different amino acid residues. To enrich the structure and function of RNA, nature modifies RNA through various chemical modifications. More than 150 structurally distinct modification types have been identified across all types of RNA [27,28]. These modifications are associated with various biological processes and human diseases [29,30]. RNA modification was initially only studied in rRNA, tRNA, and snRNA. Using immunoprecipitating RNA or covalently binding RNA-methylase complexes in combination with sequencing, the researchers gained a global understanding of the characteristics of these RNA modifications [8,31,32]. More and more RNA modifications are now confirmed, and these changes are detected in a variety of RNAs [33,34,35,36,37]. RNA modifications affect base pairing, secondary structure, and the ability of RNA to interact with proteins directly. These chemical changes further affect RNA processing, localization, translation, and decay processes to regulate gene expression [38].

### 2.1. Several Most Common RNA Modification Types

The common RNA modifications include *N*^6^ methylation of adenosine (m^6^A), *N*^1^ methylation of adenosine (m^1^A), *N*^7^-methylguanosine (m^7^G), 5-methylcytosine (m^5^C), 2′O-methylation (2′-O-Me or Nm), pseudouridine (5-ribosyl uracil or Ψ) and adenosine to inosine RNA editing (A-to-I editing), etc. (Figure 1) [39]. Among these RNA modifications, m^6^A is the most abundant form in eukaryotic cells, extensively studied in recent years. m^6^A expression was abundant in the liver, kidney, and brain. The content of it in different cancer cell lines varies greatly. Studies have found that m^6^A is mainly distributed within genes, and the proportion of m^6^A in protein-coding regions (CDS) and untranslated regions (UTRs) is relatively high. m^6^A in UTRs tended to be highly expressed in the third UTR region, while CDS regions were mainly enriched near-stop codons. The modification of m^6^A primarily occurs on the adenine of the RRACH sequence, where R is guanine, or adenine, and H is uracil, adenine, or cytosine [40]. The m^6^A modification has been implicated in the activation of multiple signaling pathways associated with lung cancer, in addition to COPD, pulmonary fibrosis, asthma, and other respiratory diseases. The m^1^A modification is formed by the methylation of *N*^1^ adenosine. It is an isoform of m^6^A and is regulated by multiple transferase complexes and demethylases, and this regulation is reversible [41,42]. The m^1^A-modified tRNA can regulate translation by increasing tRNA stability, while m^1^A-modified mRNA and lncRNA can influence RNA processing or protein translation. Some studies have found that m^1^A may regulate mitochondrial function [43]. The m^7^G was first discovered at the 5′ caps and internal positions of mRNAs, as well as inside rRNAs and tRNAs [44,45,46]. Recently, m^7^G has also been detected in miRNAs [47]. The m^7^G is associated with tumor metastasis and growth. The m^5^C is ubiquitous in mRNAs, tRNAs, RNAs, and ncRNAs [48]. It is involved in various RNA metabolisms. In tRNA, m^5^C is involved in stabilizing tRNA secondary structure and enhancing codon recognition. In addition, m^5^C modifies rRNA and ncRNA, thereby regulating mitochondrial dysfunction, stress response defects, gametocyte and embryonic development, tumorigenesis and cell migration [49,50,51]. 2′-O-Me(Nm) is a co- or post-transcriptional modification of RNA, where one of the methyl groups (-CH3) is added to the 2’ hydroxyl (-OH) of ribose. 2′-O-Me is widespread in tRNAs, RNAs, and mRNAs and regulates pre-mRNA splicing and small RNA stability [52,53]. Of all RNA species, rRNAs carry the most 2′-O-Me modifications. 2′-O-Me is found mainly in mRNA caps, but 2′-O-Me has recently been detected in CDS. 2′-O-Me can also modify sncRNAs (including miRNAs and piRNAs) [54]. Ψ can change mRNA’s secondary structure. When Ψ occurs in the stop codons, or nonsense codons it can affect the translation process and translation result [55]. Ψ could affect the development of lung cancer. A-to-I editing mainly exists on the primary transcript of mRNA, tRNAs, and miRNAs, and this RNA modification mechanism can modify the secondary structure of RNA. It is the deamination of adenosine in RNA to inosine. Inosine is recognized as guanosine in cells. A-to-I editing is associated with lung cancer cell phenotype.

### 2.2. RNA Modification Database

High-throughput sequencing data has a key impact in the study of RNA modification. These sequencing data are available on public websites, including the NCBI-Gene Expression Omnibus database (NCBI-GEO) (https://www.ncbi.nlm.nih.gov/geo, accessed on 27 November 2022). The data related to chronic lung disease in the GEO database is mainly related to m^6^A modification. The m6A levels in cisplatin-resistant A549 cells were up-regulated compared to A549 cells (GSE140020, GSE136433). In addition, multiple data sets were compared and screened for m6A markers in LUAD cells (GSE198288, GSE176348, GSE161090). There are also RNA-seq expression profiles associated with writer/eraser perturbation. For example, expression profiles after METTL3 knockdown in A549 and H1299 cells (GSE76367), ALKBH5 knockdown in PC9 cells (GSE165453), and YTHDF1 or YTHDF2 knockdown PC9 cells (GSE171634).

With the further study of RNA modification, databases containing various information are emerging. These databases, which have received widespread interest and use, in turn, provide the basis for further research about the function of RNA modification. The existing RNA modification databases can be divided into biochemical RNA modification databases, comprehensive reversible RNA modification databases, specialized reversible RNA modification databases, and RNA editing databases [56]. Biochemical RNA modification databases can query the chemical structure and biosynthetic pathways of RNA modification, among which RNA Modification Database (RNAMDB) [57] and Modomics [58] are the most common. Comprehensive reversible RNA modification databases include MethylTranscriptome DataBase (MeT-DB) [59], RNA Modification Base Database (RMBase) [60], m^6^A-Atlas [61], m^6^A2target [62], m^5^C-Atlas [63], m^7^GHub [64] and RNA Epi-transcriptome Collection (REPIC) [65]. RMBase is the most comprehensive RNA modification database available. The m^6^A2Target is a comprehensive database of target genes for m^6^A modified enzymes (writers, erasers, and readers). Specialized reversible RNA modification databases include m^6^Avar [66], m^6^A-TSHub [67], CVm^6^A [68], RMVar [69] and RMDisease [70]. RNA editing databases include RNA Editing Database (REDIdb) [71], Rigorously Annotated Database of A-to-I RNA Editing (RADAR) [72], Database of RNA Editing (DARNED) [73], and REDIportal [74]. There are other functional tools available for RNA modification such as m^6^ASNP [75] and ConsRM [76]. Genetic variants that affect RNA modification play a key role in many aspects of RNA metabolism and are also associated with chronic lung disease. It is important to assess the effect of single nucleotide variants in the human genome on m^6^A modification. The m^6^AVar database is a comprehensive database for studying m^6^A-related variants that may affect m^6^A modification. It will explain the influence of variants through the function of m^6^A modification. The m^6^AVar can be used to predict potential modification sites of m^6^A in chronic lung disease. According to the m^6^AVar database, three m^6^A-related genes (ZCRB1, ADH1C, and YTHDC2) are reliable prognostic indicators for lung adenocarcinoma (LUAD) patients and are potential therapeutic targets [77]. RMVar is similar to m^6^AVar in that it mainly collects related variants affecting m^6^A modification. However, RMVar has more comprehensive functions than m6AVar. RMVar also contains other RNA-modified variants. The databases of genetic variation in RNA modification also include RMDisease and m6A- TSHub. RMDisease integrates the predictions of 18 different RNA modification prediction tools and a large number of experimentally validated RNA modification sites and identifies single nucleotide polymorphisms (SNPs) that may affect eight types of RNA modification. Most of the m^6^A-associated cancer variants are tissue- and cancer-specific. The m^6^A-TSHub consists of four core components, namely m^6^A-TSDB, m^6^A-TSFinde, m^6^A-TSVar, and m^6^A-CAVar. The m^6^A-TSDB platform can be used to retrieve m^6^A sites in normal lung tissue and lung cancer cell lines. Then, m^6^A-TSVa can be used to explore the influence of lung tissue variation on m^6^A by integrating tissue-specific m^6^A. Finally, m^6^A-CAVar can be used to screen for cancer variants affecting m^6^A in lung tissue. Through these databases, we can screen out the gene variants associated with RNA modification in chronic lung disease for further verification.

### 2.3. The Regulation of RNA Modification

Three proteins have been found to regulate RNA modification. The first is an enzyme that introduces modified nucleotides into RNA during post-transcriptional RNA modification; the second protein interacts with the modified nucleotides; the third protein removes the modification labels [78]. The methylation modification of m^6^A is regulated by three types of proteases, methyltransferases (writers, including methyltransferase like protein-3/14/16 (METTL3/14/16), RNA-binding motif protein 15/15B (RBM15/15B), zinc finger CCCH type containing 13 (ZC3H13), virlike m^6^A methyltransferase associated (VIRMA, also known as KIAA1429), cbl proto-oncogene like 1 (CBLL1), and Wilms’ tumor-associated protein (WTAP), demethylases (erasers, including fat mass and obesity-associated (FTO) and alkB homolog 5 (ALKBH5)), and readers (including YTH domain family 1/2/3 (YTHDF1/2/3), YTH domain containing 1/2 (YTHDC1/2), insulinlike growth factor 2 mRNA binding protein 1/2/3 (IGF2BP1/2/3), and heterogeneous nuclear ribonucleoprotein A2B1 (HNRNPA2B1)), and is reversible and can be dynamically regulated [79,80]. These regulatory proteases work together in a coordinated manner to maintain a homeostatic balance of intracellular m^6^A levels. Reversible m^1^A methylomes are achieved by the dynamic modulation of m^1^A RNA-modifying proteins, including m^1^A methyltransferases such as tRNA methyltransferase 6 noncatalytic subunit (TRMT6)-TRMT61A complex, TRMT10C, TRMT61B and nucleomethylin (NML), m^1^A demethylases such as ALKBH1, ALKBH3 and FTO, and m^1^A-dependent RNA-binding proteins such as YTHDF1/2/3 and YTHDC1 [81]. The modification of m^7^G in mammals is catalyzed by the compounds METTL1 and WD repeat domain 4 (WDR4), a complex that facilitates the installation of m^7^G in tRNA, miRNA, and mRNA [82,83]. RNA guanine-7 methyltransferase (RNMT) and its cofactor RNMT-activating miniprotein (RAM) actively catalyze m^7^G. Among them, RNMT is the catalytic subunit, and RAM is the regulatory subunit, which plays the role of activation. Williams–Beuren syndrome chromosome region 22 (WBSCR22) and TRMT112 are responsible for regulating m^7^G in rRNA. The main function of these regulatory mechanisms is to add m^7^G to the target RNAs, thereby mediating many key biological processes by modulating RNA production, structure, and maturation [84]. The m^5^C is reversibly regulated by methyltransferases, including DNA methyltransferase (DNMTs, such as DNMT1, DNMT2, and DNMT3A/3B) and NOP2/Sun RNA methyltransferases (NSUNs), and demethylases, including ten-eleven translocation (TETs), and reader proteins including YTHDF2, Aly/REF Export Factor(ALYREF) and Y-box binding protein 1(YBX1) [85,86]. There are two ways to add 2′-O-Me modification: either by the complex assembly of proteins associated with snoRNA guides (sno(s)RNPs) to carry out site-specific modification or standalone protein enzymes with direct specific site modification [87,88]. In addition, the methylated reader protein TAR RNA-binding protein (TRBP) binds to the methyltransferase FtsJ RNA 2’-O-methyltransferase 3 (FTSJ3) to form a TRBP-FTSJ3 complex, which induces 2′-O-Me [89]. Fibrillarin (FBL) is also a 2’-O-methyltransferase that can form a small nucleolar ribonucleoproteins (snoRNPs) with three other proteins and snoRNA for specific rRNA modifications [90]. Ψ is produced by the isomerization of uridine, catalyzed by pseudouridine synthase (PUS). Thirteen pseudouridine synthases have been identified, which can be divided into two categories, RNA-dependent and RNA-independent PUSs. Dyskerin pseudouridine synthase 1 (DKC1) is the catalytic subunit of the H/ACA snoRNP complex and catalyzes rRNA pseudouridylation. The other 12 writers are PUSs: PUS1, PUSL1, PUS3, TRUB1, TRUB2, PUS7, PUS7L, RPUSD1-4, and PUS10. The cellular localization and RNA targets of these enzymes are fixed. No Ψ erasers and readers have been identified. This is probably because of the formation of relatively inert C-C bonds between the ribose and the base, which leads to the fact that the pseudoureacylation process is irreversible [91]. A-to-I editing is catalyzed by adenosine deaminases acting on the RNA (ADAR) family of enzymes. There are 3 ADAR enzymes, ADAR1 and ADAR2 being catalytically active, while ADAR3 lacks catalytic activity [92].

## 3. RNA Modifications in Lung Cancer and Other Chronic Lung Diseases

### 3.1. The Roles of RNA Modifications

RNA modifications have a key influence in the regulation of many fundamental biological processes associated with chronic lung disease (Table 1). The methylation of m^6^A can affect the RNA stability, localization, turnover, and translation efficiency of genes, thereby regulating cellular processes such as cell self-renewal, differentiation, invasion, and apoptosis, and is even critical for skeletal development and homeostasis [93,94]. The METTL3-METTL14 complex increases the expression of the cyclin-dependent kinase p21 by regulating m^6^A. This complex enhances m^5^C methylation, which synergistically promotes p21 expression and affects oxidative stress-induced cellular senescence [95]. In the cardiovascular system, multiple m^6^A-related regulators promote the progression of atherosclerosis by regulating macrophage polarization and inflammation. WTAP and METTL14 also can affect the phenotypic regulation of vascular smooth muscle cells (VSMCs) via m^6^A modification [96]. Enhanced m^6^A RNA methylation generates compensatory cardiac hypertrophy, whereas decreased m^6^A causes cardiomyocyte remodeling and dysfunction [97]. An increasing number of studies have found that m^6^A modification plays an important role in controlling the generation and self-renewal of hematopoietic stem cells and in mediating the development and differentiation of T and B lymphocytes from hematopoietic stem cells [98]. YTHDF2, the first recognized “reader” of m^6^A, can maintain the homeostasis and maturation of natural killer (NK) cells, and positively regulate the antitumor and antiviral activities of NK cells. Its deletion significantly impairs NK cell antitumor and antiviral activity in vivo [99]. METTL3-mediated m^6^A methylation is also associated with NK cell homeostasis and antitumor immunity [100]. In addition, m^6^A is associated with apoptosis, autophagy, pyroptosis, ferroptosis, and necrosis [101], and m^5^C mediates cell proliferation, differentiation, apoptosis, and stress response [102].

### 3.2. RNA Modifications in the Respiratory System

RNA epigenetic modifications have been broadly reported in lung cancer development and other chronic lung diseases (Figure 2). The most studied m^6^A modification plays an important role in tumorigenesis, proliferation, and metastasis. Active m^6^A regulators in lung cancer are related with the activation of multiple signaling pathways, such as DNA replication, RNA metabolism, epithelial–mesenchymal transition (EMT), cell cycle, cell proliferation and apoptosis, energy metabolism, inflammatory response, drug resistance, tumor metastasis and recurrence [103]. Reprogramming of energy metabolism is the hallmarks of cancer. The m^6^A regulates tumor metabolism by directly regulating nutrient transporters and metabolic enzymes or indirectly by controlling key components in metabolic pathways [104]. Aberrant m^6^A modification contributes to the progression of malignant tumors and affects patient prognosis. Some m^6^A-related mRNA markers have also been shown to be independent prognostic biomarkers in patients with different types of cancer [105,106]. Recent studies have found that the m^6^A methylase METTL3 is abnormally activated in cisplatin-resistant nonsmall-cell lung cancer (NSCLC) cells. METTL3 enhances *YAP* mRNA translation by introducing YTHDF1/3 and eIF3b into the translation initiation complex and increasing *YAP* mRNA stability by regulating the MALAT1-miR-1914-3p-YAP axis. This induces NSCLC drug resistance and metastasis [107]. FTO is the first m^6^A demethylase discovered to promote NSCLC cell growth by demethylating *USP7* mRNA or *MZF1* mRNA transcripts and increasing their stability and transcriptional levels [108,109]. The 5’-UTR of m^6^A-modified *PDK4* mRNA positively regulates the glycolysis of cells by binding to IGF2BP3, thereby promoting the development of cancer. While METTL14 knockout can reverse the function of IGF2BP3 and play a tumor suppressor role [110]. In addition, IGF2BP1 can also promote cancer development and induce therapeutic resistance by stabilizing oncogenic mRNAs [111]. The m^6^A modification also affects the mediating tumor metastasis process. When lung cancer brain metastasis (BM) occurs, miR-143-3p was upregulated in paired BM tissues compared with primary lung cancer tissues. METTL3-mediated m^6^A modification induces the maturation of miR-143-3p, which induces lung cancer invasion and angiogenesis via suppressing the expression of the target gene vasohibin-1 [112]. The lncRNA HCG11 is modulated by METTL14-mediated m^6^A modification in LUAD. METTL14-mediated HCG11 inhibits the growth of LUAD by targeting large tumor suppressor kinase 1 (*LATS1*) mRNA through IGF2BP2 [113]. The m^7^G methyltransferase METTL1 and WDR4 complex is significantly elevated in lung cancer, which can promote lung cancer cell growth and invasion and negatively correlate with patient prognosis. Impaired m^7^G tRNA modification in the absence of METTL1/WDR4 results in decreased proliferation, colony formation, cell invasion, and tumorigenicity of lung cancer cells [114]. Highly expressed METTL1 can also methylate mature let-7 miRNAs by interfering with inhibitory secondary structure (G-quadruplex) in the pri-miRNA transcript of let-7. The m^7^G-let-7 miRNA can inhibit lung cancer cell metastasis by reducing the expression of target oncogenes, including high mobility group AT-hook 2 (*HMGA2*), *RAS*, and *MYC*, at the posttranscriptional level [47]. Studies have confirmed that m^5^C levels can be used as a cancer marker. For example, in lung squamous cell carcinoma (LUSC), the upregulation of m^5^C-related NSUN3 and NSUN4 is related to poor patient prognosis [115]. In LUAD, cells with high NSUN1 expression are more likely to be poorly differentiated [116]. Abnormally elevated RNA m^5^C levels can be found in circulating tumor cells from lung cancer patients [117]. The studies of Ψ are mainly concerned with breast, lung, and prostate cancers. In NSCLC, the expression of lncRNAs PCAT1 is highly expressed, and cooperates with DKC1 to influence the proliferation, invasion, and apoptosis of NSCLC cells through the VEGF/AKT/Bcl-2/caspase9 pathway [118]. The rs9309336 may interfere with PUS10 expression, thereby reducing the sensitivity of tumor cells to tumor necrosis factor-associated apoptosis-inducing ligand (TRAIL) [119]. Finally, it promotes the immortalization of tumor cells and the development of lung cancer. The RNA editing protein ADAR promotes LUAD progression by stabilizing transcripts encoding focal adhesion kinase (FAK). The increased abundance of ADAR in the mRNA and protein levels in lung tissues of LUAD patients was associated with tumor recurrence. ADAR increases the stability of *FAK* mRNA by binding to FAK. FAK blocks the ADAR-induced invasiveness of LUAD cells [120]. A-to-I microRNA editing is correlated with tumor phenotypes in multiple cancer types. Altered editing levels of microRNAs in LUAD may be a potential biomarker [121].

**Figure 2 genes-13-02381-f002:**
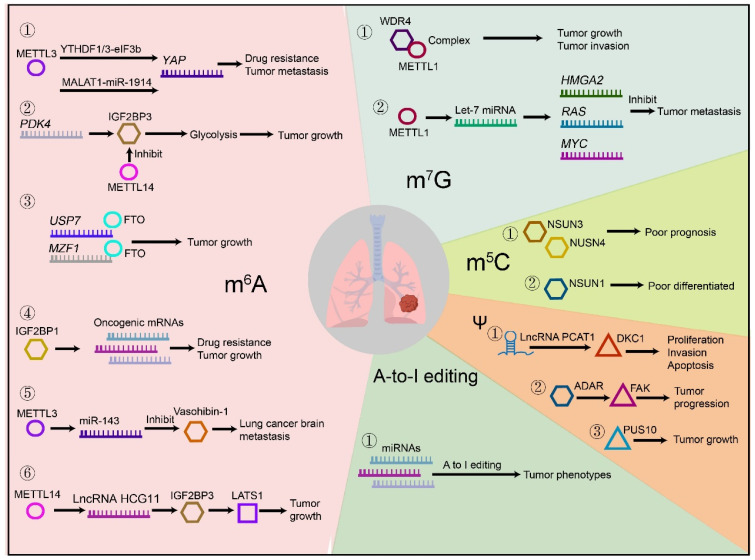
The mechanisms and pathways of RNA modifications in lung cancer. The RNA modifications that have been confirmed to be involved in the development and progression of lung cancer include m^6^A, m^7^G, m^5^C, Ψ, and A to I editing. The most studied modification in lung cancer is m^6^A. METTL3 increases the expression of *YAP* mRNA by combining YTHDF1/3-eIF3b axis and MALAT1-miR-1914 axis, causing drug resistance and metastasis of lung cancer. The m^6^A-modified *PDK4* mRNA enhances cellular glycolysis through IGF2BP3, thereby promoting tumor growth. The knockdown of *METTL14* reversed this process. *USP7* mRNA or *MZF1* mRNA demethylated by FTO can promote the tumor growth. IGF2BP1 can promote drug resistance and tumor growth by stabilizing oncogenic mRNAs. METTL3 induces the maturation of miR-143 and inhibits the expression of Vasohibin-1, thereby inducing lung cancer brain metastasis. METTL14 regulates the expression of lncRNA HCG11 and then inhibits tumor growth by targeting *LATS1* mRNA via IGF2BP2. The m^7^G methyltransferase METTL1 and WDR4 complex promotes the tumor growth and invasion. In addition, METTTL1 promotes the methylation of Let-7 miRNA, increases the expression of *HMGA2*, *RAS* and *MYC*, and induces the tumor metastasis. The m^5^C-associated upregulations of NSUN3 and NSUN4 are associated with poor prognosis. Cancer cells with high NSUN1 expression are related with poor differentiated. LncRNA PCAT1 and Ψ methyltransferase DKC1 cooperate to promote the proliferation, invasion and apoptosis of lung cancer cells. ADAR promotes tumor progression by stabilizing FAK transcripts. The expression of Ψ methyltransferase PUS10 is associated with the tumor growth. A-to-I miRNA editing correlates with tumor phenotype.

**Table 1 genes-13-02381-t001:** A summary of RNA modifications in chronic lung diseases.

Modification Type	Writer	Eraser	Reader	Mechanism	Reference
m6A	METTL3/14/16, RBM15/15B, ZC3H3, VIRMA, CBLL1, WTAP, KIAA1429	FTO, ALKBH5	YTHDF1/2/3, YTHDC1/2, IGF2BP1/2/3, HNRNPA2B1	(1) Activation of METTL3 promotes drug resistance and metastasis in NSCLC; (2) FTO promotes the growth of NSCLC cells by demethylating *USP7* mRNA or *MZF1* mRNA transcripts; (3) *PDK4* mRNA binds to IGF2BP3 and promotes the development of cancer. METTL14 knockout can reverse the function of IGF2BP3; (4) IGF2BP1 promotes cancer development and induces therapeutic resistance; (5) METTL3 inhibited vasohibin-1 expression and induced lung cancer invasion and angiogenesis; (6) METTL14 inhibits LUAD growth through IGF2BP2; (7) High expression m6A promotes the development and progression of COPD and participates in PM2.5-induced microvascular injury; (8) Inhibition of METTL3 function promotes LPS-induced pneumonia; (9) METTL3 promotes pulmonary fibrosis; (10) ALKBH5 promotes silicon-induced pulmonary fibrosis; (11) YTHDF3 affects eosinophil function in severe asthma.	[85,86,87,88,89,90,91,103,104,106,107,108,109,110]
m1A	TRMT6-TRMT61A complex, TRMT10C, TRMT61B, NML	ALKBH1/3, FTO	YTHDF1-3, YTHDC1		
m7G	METTL1 and WDR4 complex, RNMT and RAM complex, WBSCR22, TRMT112			(1) METTL1 and WDR4 complexes promote lung cancer cell growth and invasion; (2) High expression of METTL1 inhibits metastasis of lung cancer cells; (3) The m7G has a key value in the prognosis and early diagnosis of IPF patients.	[47,92,111]
m5C	NSUNs, DNMT1, DNMT2, DNMT3A/3B	TETs	ALYREF, YBX1, YTHDF2	(1) Upregulation of NSUN3 and NSUN4 is associated with poor prognosis in LUSC patients; (2) Cells with high NSUN1 expression are more likely to be poorly differentiated in LUAD; (3) Increased m5C modification negatively affects normal lung metabolic activities.	[93,94,105]
2′-O-Me	FBL, FTSJ3		TRBP		
Ψ	DKC1, PUS1, PUSL1, PUS3, TRUB1, TRUB2, PUS7, PUS7L, RPUSD1-4, PUS10			(1) LncRNAs PCAT1 is highly expressed in NSCLC and cooperates with DKC1 to affect proliferation, invasion and apoptosis of NSCLC cells; (2) PUS10 promotes the immortalization of tumor cells and the development of lung cancer; (3) ADAR promotes LUAD progression.	[96,97,98]
A-to-I editing	ADAR1, ADAR2			(1) A-to-I microRNA editing is correlated with tumor phenotypes.	[99]

In addition to studies on lung cancer, the effects of RNA modification in other chronic lung diseases such as COPD, pneumonia, asthma, and pulmonary fibrosis have also been explored (Figure 3). Exposure to some toxicants can cause A-to-I editing of lung cells. Potassium chromate (VI) induced upregulation of ADARB1 in human lung cells [122], whereas tetrachlorodibenzodioxin (TCDD) exposure resulted in the decreased expression of ADARB1 [123]. Carbon nanotubes can increase ADAR expression in mouse lungs [124]. It has been found that the expression of m^6^A RNA methylation regulators is abnormal in COPD, among which the mRNA expressions of IGF2BP3, FTO, METTL3, and YTHDC2 show a tight association with the occurrence of COPD. IGF2BP3, FTO, METTL3, and YTHDC2 have obvious correlations with various important genes enriched in signaling pathways and biological processes that promote the development and progression of COPD [125]. Exposure to fine particulate matter (PM_2.5_) is an important cause of COPD. METTL16 may regulate sulfate expression through m^6^A modification, thereby participating in PM_2.5_-induced microvascular injury and promoting the development of COPD [126]. In addition, the increase of mRNA m^5^C modification may negatively affect normal lung metabolic activities by upregulating gene expression levels in the lungs of mice exposed to PM_2.5_ [127]. LncRNA small nucleolar RNA host gene 4 (SNHG4) promotes LPS-induced lung inflammation by inhibiting METTL3-mediated expression of *STAT2* mRNA m^6^A [128]. Myofibroblasts are the main collagen-producing cells in pulmonary fibrosis, which are mostly derived from resident fibroblasts via fibroblast-to-myofibroblast transition (FMT). m^6^A modification was upregulated in the bleomycin (BLM)-induced pulmonary fibrosis mouse model, FMT-derived myofibroblasts, and lung samples from IPF patients. Silencing METTL3 can inhibit FMT by reducing m^6^A levels. KCNH6 is involved in the m^6^A-regulated FMT process. m^6^A modification regulates KCNH6 expression through YTHDF1 [129,130]. ALKBH5 promotes silica-induced pulmonary fibrosis through miR-320a-3p/forkhead box protein M1 (FOXM1) axis or directly targeting FOXM1. Targeting ALKBH5 can be used to treat pulmonary fibrosis [131]. The m^6^A methylation has also been implicated in the pathogenesis of asthma, and YTHDF3 has an effect on eosinophils for severe asthma, which can guide future immunotherapy strategies [132]. Increased METTL1 level in IPF patients is associated with poor prognosis. IPF can be divided into two molecular subtypes (subtype 1 and subtype 2) by combining the expression levels of METTL1 and RNMT. Patients with subtype 2 have a more unfavorable prognosis than patients with subtype 1. It suggests that m^7^G has an important value in predicting the prognosis of IPF patients and early diagnosis of IPF patients [133].

## 4. Diagnosis and Therapeutic Potential

In recent years, RNA modification has been identified as a novel regulatory mechanism in controlling cancer pathogenesis and treatment response/resistance. The m^6^A modification plays a multifunctional role in normal and abnormal biological processes, and its regulatory proteins can act as therapeutic targets for cancer and are expected to be biomarkers for overcoming drug resistance [134]. METTL3 is the major catalytic subunit of m^6^A modification. METTL3 facilitates the translation of a large subset of oncogenic mRNAs and has direct physical and functional interactions with translation initiation factor 3 subunit h (eIF3h). METTL3-eIF3h interaction is required for oncogenic transformation. The depletion of METTL3 inhibits tumorigenicity and sensitizes lung cancer cells to bromodomain-containing protein 4 (BRD4) inhibition [135]. METTL3 promotes tumor development in human lung cancer cells by upregulating the translation of important oncogenes such as EGFR and TAZ. MiR-33a, a negative regulator of METTL3, can directly target the 3’UTR of *METTL3* mRNA, reduce its expression, and further inhibit NSCLC cell proliferation [136]. The dynamic m^6^A methylome is a new mechanism for drug resistance in cancer, such as tyrosine kinase inhibitors (TKIs) [137]. FTO is an oncogene of LUSC, and its increased expression can promote the growth of cancer cells. Knockout of FTO significantly decreased *MZF1* mRNA level, and MZF1 gene silencing significantly inhibited the cell viability and invasion of LUSC [109]. Two FTO inhibitors, FB23 and FB23-2, can attenuate the activity of FTO demethylase by directly binding to the activity pocket of FTO demethylase, resulting in a significant lethal effect on cancer cells [138]. Under the action of intermittent hypoxia (IH), the expression of ALKBH5 was upregulated in lung cancer cells, resulting in a decrease in the level of m^6^A. Knockdown of *ALKBH5* under this condition significantly inhibited cell invasion by upregulating the m^6^A level of *FOXM1* mRNA and reducing its translation efficiency [139]. Downregulation of solute carrier 7A11 (SLC7A11) expression in lung cancer inhibits cell proliferation and colony formation. In LUAD, m^6^A modification destabilizes *SLC7A11* mRNA and accelerates mRNA decay upon recognition by YTHDC2, thereby affecting cystine uptake and contributing to antitumor activity [140]. The m^6^A modification-related inflammatory cytokine interleukin 37 (IL-37) has received extensive attention for the treatment of lung cancer. IL-37 inhibits the proliferative capacity of LUAD cells by regulating RNA methylation. Meanwhile, overexpression of IL-37 decreased the expression of ALKBH5 and thus can also be used to treat NSCLC patients [141]. Gefitinib is indicated for the treatment of locally advanced or metastatic NSCLC. However, acquired resistance limits its long-term efficacy. m^6^A modification reduces gefitinib resistance (GR) in NSCLC patients via the FTO/YTHDF2/ABCC10 axis [142]. The development of selective inhibitors to RNA modification regulators for future clinical applications would create more effective therapeutic approaches for treating lung cancers and chronic lung diseases.

## 5. Conclusions and Future Perspectives

In conclusion, this article reviews the recent advances in the function and molecular mechanism of RNA epigenetics in the progression of chronic lung diseases. RNA epigenetics is expected to be a research tool for the development of new diagnostic biomarkers with clinical value. Enzymes involved in regulating RNA modification can be new targets for the treatment of chronic lung diseases. Most of the m^6^A regulators are upregulated in cancer and play a role in promoting tumor growth. These regulators, including METTL3, METTL14, and WTAP, and their key targets are associated with the clinical characteristics of various cancer patients, which may provide new possibilities for early cancer diagnosis [143]. The combination of m^6^A targeted drugs with traditional chemotherapy drugs or PD-1/PD-L1 inhibitors has great therapeutic prospects [144]. The mechanism by which IL-37 inhibits the proliferation of LUAD cells and is used to treat NSCLC patients is also related to m^6^A methylation [141]. YTHDF1 knockout can significantly enhance the therapeutic effect of PD-L1 immune checkpoint blocking, suggesting that YTHDF1 is a potential therapeutic target in tumor immunotherapy [145]. The m^5^C deletion of mitochondrial RNA in tumor cells can reduce the metastasis and invasion of cancer cells. This means that when cancer patients undergo clinical treatment, they can inhibit the metastasis and spread of cancer cells by inhibiting m^5^C modification in mitochondria, thus increasing clinical benefits. As an enzyme responsible for modifying RNA, NSUN3 is only used to modify the formation of m^5^C, so it is a very promising drug target [146]. RNA modification provides a new research direction for the early diagnosis and treatment of tumors.

In the future, precision medicine based on RNA epigenetics may target individual patients with chronic lung diseases for diagnosis and treatment. However, the function of these chemical modifications in both coding and noncoding RNAs remains in its infancy, and collaborative efforts are still needed to establish a clear link between RNA epigenetics and chronic lung diseases. At present, many new methods of RNA modification have emerged, among which Direct RNA sequencing is the representative one. Through the unique way of the library building and relying on the algorithm, the collection of RNA modification information at the single base level can be achieved. The development of RNA modification quantitative map databases (such as DirectRMDB) based on this sequencing method also undoubtedly opens another direction for RNA modification detection [147]. In brief, the advances in RNA epigenetics detection technologies will undoubtedly lead to the discovery of new mechanisms regulating gene expression in chronic lung diseases in the future.

## Figures and Tables

**Figure 1 genes-13-02381-f001:**
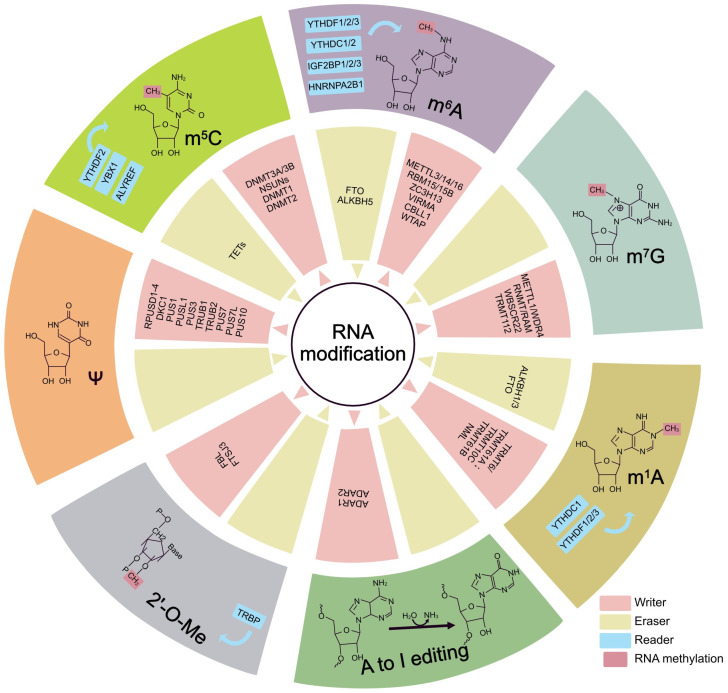
The most common types of RNA modifications. Common RNA modifications include *N*^6^ methylation of adenosine (m^6^A), *N*^1^ methylation of adenosine (m^1^A), *N*^7^-methylguanosine (m^7^G), 5-methylcytosine (m^5^C), 2′O-methylation (2′-O-Me), pseudouridine (Ψ), and adenosine to inosine RNA editing (A-to-I editing), and regulated by methyltransferase (writers), demethylases (erasers), and some specific proteins (readers).

**Figure 3 genes-13-02381-f003:**
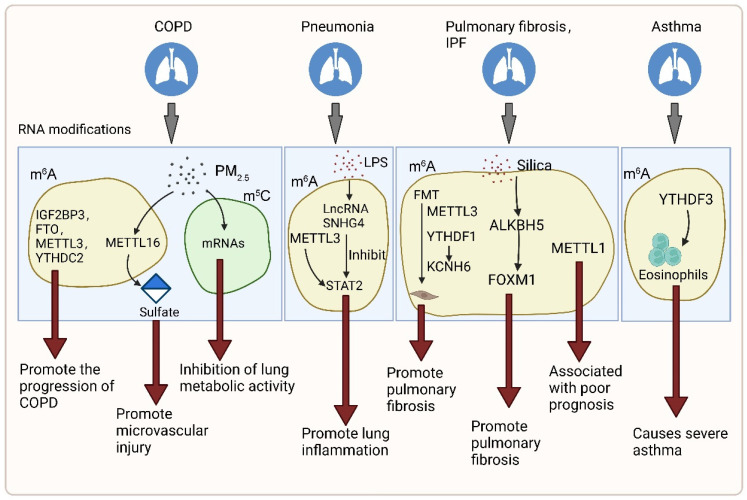
The mechanisms and pathways of RNA modifications in chronic lung diseases. RNA modifications promote the occurrence and development of chronic lung diseases such as COPD, pneumonia, asthma, and pulmonary fibrosis. Among these modifications, m^6^A have been the most studied in chronic lung diseases. The mRNA expressions of m^6^A related IGF2BP3, FTO, METTL3 and YTHDC2 promote the progression of COPD. The m^6^A methyltransferase METTL16 regulates the level of sulfate and participates in microvascular injury induced by PM_2.5_. In addition, increased m^5^C modification is also involved in the process of PM_2.5_ induced COPD by inhibiting lung metabolic activity. LncRNA SNHG4 promotes LPS-induced pneumonia by inhibiting METTL3-mediated *STAT2* mRNA expression. The enhancement of FMT promotes pulmonary fibrosis. METTL3 increases the FMT process. The m^6^A modification also is involved in FMT process by regulating KCNH6 expression through YTHDF1. METTL1 level in IPF patients is positive associated with poor prognosis. ALKBH5 promotes silicon-induced pulmonary fibrosis through FOXM1. The m^6^A-related YTHDF3 causes severe asthma by affecting eosinophils.

## Data Availability

Not applicable.

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
