# Peer review of "RNA Epigenetics in Chronic Lung Diseases"

_genes, 2022, doi:10.3390/genes13122381_

Round 1
Reviewer 1 Report
In this manuscript, the authors reviewed the studies of RNA epigenetics in chronic lung disease. The information about different RNA modifications such as m6A, m7G and m5C was summarized. The relationship between these modifications and chronic lung disease was also reviewed. Here are my concerns:
(1) The organization is not clear according to the section titles. It seems there are only two sections of the manuscript, which are “Introduction” and “Conclusions and future perspectives”.
(2) Is there any prior reviews about RNA epigenetics and lung diseases? If yes, the authors should first introduce these works and then state the differences between this manuscript and previous reviews.
(3) It would be clear if the authors could review the relationship between different modifications with lung diseases in different sections or paragraphs.
(4) Where is the Table 6?
(5) Is it true that “There are three proteins involved in RNA modification”?
(6) The authors stated “The methylation modification of m6A is regulated by three types of proteases,”, which three types of proteases are involved in the modification of m6A. According to my knowledge, proteases are a kind of enzyme which was used to hydrolyze proteins.
(7) The language should be polished.
Author Response
In this manuscript, the authors reviewed the studies of RNA epigenetics in chronic lung disease. The information about different RNA modifications such as m6A, m7G and m5C was summarized. The relationship between these modifications and chronic lung disease was also reviewed. Here are my concerns:
(1) The organization is not clear according to the section titles. It seems there are only two sections of the manuscript, which are “Introduction” and “Conclusions and future perspectives”.
Response: We thank the reviewer for the suggestion. We have reworked the organization of our manuscript to divide into five sections. In addition, we divided the second and third sections of our manuscript into three and two parts respectively.
(2) Is there any prior reviews about RNA epigenetics and lung diseases? If yes, the authors should first introduce these works and then state the differences between this manuscript and previous reviews.
Response: We thank the reviewer for the suggestion. For the answer to this question, we have added it in line 66-79 on page 3 (in red).
(3) It would be clear if the authors could review the relationship between different modifications with lung diseases in different sections or paragraphs.
Response: We thank the reviewer for the suggestion. For the answer to this question, we have added some content on line 129-132, line140-141 on page 5, 157, 161-162 on page 6 (in red).
(4) Where is the Table 6?
Response: We thank the reviewer for the suggestion. However, we did not find any words about the “table6” in our manuscript. We would be grateful if you could point it out.
(5) Is it true that “There are three proteins involved in RNA modification”?
Response: We thank the reviewer for the suggestion. We reviewed the literature and confirmed that similar conclusions appeared in other literature. 1) ”RNA modifications are dynamically mediated by three different classes of proteins: writers, erasers and readers (WERs)”(PMID: 36300630). 2) “Epigenetic marks are mediated by three different
classes of proteins: writers, erasers and readers”(PMID: 32300195).
(6) The authors stated “The methylation modification of m6A is regulated by three types of proteases,”, which three types of proteases are involved in the modification of m6A. According to my knowledge, proteases are a kind of enzyme which was used to hydrolyze proteins.
Response: We thank the reviewer for the suggestion. We have reviewed the literature, and we think it should be more accurate to change to“The methylation modification of m6A is regulated by three types of protein” here. Thank you for helping us find this mistake.
(7) The language should be polished.
Response: We thank the reviewer for the suggestion. We have polished the language.
Reviewer 2 Report
In this manuscript, Yan et al. summarized studies on the roles of several prevalent RNA modification in chronic lung diseases. While the topic of this review is interesting, there are several major scientific issues that should be addressed.
1. The authors should seriously check their annotation of writer/reader/erasers. For example, FBL is missed as a potential 2’-O-Me writer, while the citation supporting YTHDF2 as an m5C reader is wrong and the YTHDC1/2 are wrongly spelled in Figure 1.
2. The authors should also carefully check the modification names and the chemical structures. For example, 2’-OMe should be 2’-O-Me, and the chemical structure of pseudouridine is not correct. The authors are also encouraged to double-check on RNA modification database like MODOMICS for RNA modification nomenclature, related enzymes and chemical structures.
3. The inclusion of pneumonia, especially LPS-induced pneumonia (which is often acute but not chronic) in the scope of chronic lung diseases is questionable. The authors should elaborate more on why it is reasonable to include such studies.
4. For the studies of RNA modification regulations, high-throughput sequencing data play a critical, and sometimes irreplaceable role. Therefore, the authors are highly recommended to summarize the high-throughput sequencing data for identifying modification in chronic-lung-disease-related samples (for example, m6A profiles in lung cancer cell line A549) and/or RNA-seq expression profile related to writer/eraser perturbation in such samples (for example, expression profiles after METTL3 knockdown in A549). Such information can be easily queried from the NCBI GEO database.
Minor points:
1. Although the overall language quality of this manuscript is clearly acceptable, the authors should take particular care of grammatical errors in long sentences (for example, line 53-57). I would rather like to suggest split such sentences into a few short sentences.
2. Some section titles are wrongly numbered (for example, lines 329 and 367).
3. The authors are encouraged to elaborate more to discuss future disease mechanism studies and clinical application in the final Conclusions and future perspectives.
Author Response
In this manuscript, Yan et al. summarized studies on the roles of several prevalent RNA modification in chronic lung diseases. While the topic of this review is interesting, there are several major scientific issues that should be addressed.
- The authors should seriously check their annotation of writer/reader/erasers. For example, FBL is missed as a potential 2’-O-Me writer, while the citation supporting YTHDF2 as an m5C reader is wrong and the YTHDC1/2 are wrongly spelled in Figure 1.
Response: We thank the reviewer for the suggestion. We have corrected these errors.
- The authors should also carefully check the modification names and the chemical structures. For example, 2’-OMe should be 2’-O-Me, and the chemical structure of pseudouridine is not correct. The authors are also encouraged to double-check on RNA modification database like MODOMICS for RNA modification nomenclature, related enzymes and chemical structures.
Response: We thank the reviewer for the suggestion. We have corrected these errors.
- The inclusion of pneumonia, especially LPS-induced pneumonia (which is often acute but not chronic) in the scope of chronic lung diseases is questionable. The authors should elaborate more on why it is reasonable to include such studies.
Response: We thank the reviewer for the suggestion. We have found in some studies that researchers use LPS to model chronic lung diseases. The references are as follows.
1: Snyder JC, Reynolds SD, Hollingsworth JW, Li Z, Kaminski N, Stripp BR. Clara cells attenuate the inflammatory response through regulation of macrophage behavior. Am J Respir Cell Mol Biol. 2010 Feb;42(2):161-71. doi:10.1165/rcmb.2008-0353OC.
2: Sagiv A, Bar-Shai A, Levi N, Hatzav M, Zada L, Ovadya Y, Roitman L, Manella G, Regev O, Majewska J, Vadai E, Eilam R, Feigelson SW, Tsoory M, Tauc M, Alon R, Krizhanovsky V. p53 in Bronchial Club Cells Facilitates Chronic Lung Inflammation by Promoting Senescence. Cell Rep. 2018 Mar 27;22(13):3468-3479. doi: 10.1016/j.celrep.2018.03.009.
3: Mao K, Luo P, Geng W, Xu J, Liao Y, Zhong H, Ma P, Tan Q, Xia H, Duan L, Song S, Long D, Liu Y, Yang T, Wu Y, Jin Y. An Integrative Transcriptomic and Metabolomic Study Revealed That Melatonin Plays a Protective Role in Chronic Lung Inflammation by Reducing Necroptosis. Front Immunol. 2021 May 4;12:668002. doi: 10.3389/fimmu.2021.668002.
4: Vernooy JH, Dentener MA, van Suylen RJ, Buurman WA, Wouters EF. Long-term intratracheal lipopolysaccharide exposure in mice results in chronic lung inflammation and persistent pathology. Am J Respir Cell Mol Biol. 2002 Jan;26(1):152-9. doi: 10.1165/ajrcmb.26.1.4652.
5: Yadava K, Pattaroni C, Sichelstiel AK, Trompette A, Gollwitzer ES, Salami O, von Garnier C, Nicod LP, Marsland BJ. Microbiota Promotes Chronic Pulmonary Inflammation by Enhancing IL-17A and Autoantibodies. Am J Respir Crit Care Med. 2016 May 1;193(9):975-87. doi: 10.1164/rccm.201504-0779OC.
- For the studies of RNA modification regulations, high-throughput sequencing data play a critical, and sometimes irreplaceable role. Therefore, the authors are highly recommended to summarize the high-throughput sequencing data for identifying modification in chronic-lung-disease-related samples (for example, m6A profiles in lung cancer cell line A549) and/or RNA-seq expression profile related to writer/eraser perturbation in such samples (for example, expression profiles after METTL3 knockdown in A549). Such information can be easily queried from the NCBI GEO database.
Response: We thank the reviewer for the suggestion. For the answer to this question, we have added it in line 164-176 on page 5-6 (in red).
Minor points:
- Although the overall language quality of this manuscript is clearly acceptable, the authors should take particular care of grammatical errors in long sentences (for example, line 53-57). I would rather like to suggest split such sentences into a few short sentences.
Response: We thank the reviewer for the suggestion. We have corrected the sentence.
- Some section titles are wrongly numbered (for example, lines 329 and 367).
Response: We thank the reviewer for the suggestion. We have corrected these errors.
- The authors are encouraged to elaborate more to discuss future disease mechanism studies and clinical application in the final Conclusions and future perspectives.
Response: We thank the reviewer for the suggestion. For the answer to this question, we have added it in the final Conclusions and future perspectives (in red).
Reviewer 3 Report
In this presented review concerning RNA epitranscriptome in chronic lung disease, Yan et al., systematically reviewed/introduced the background/biological functions/lung disease involvement of seven well-studied RNA modifications, namely, m6A, A-to-I, m1A, m5C, Psi, m7G, and Nm. In general, the review is well-organized with serval sub-section introducing WER (writer, eraser, reader) and mechanisms/pathways of each individual modification type. Nevertheless, besides the wet-lab approaches for unveiling disease associations, many well-known computational methods have also been developed to study RNA modification in lung disease, including lung cancer. I have the following comments and hope to further improve the review quality.
1, In the manuscript, the Figures are very clear and informative. It would be even better to include some Tables for summarizing/concluding the review finding. Tables are very informative/useful in REVIEW article.
2, For INTRODUCTION section, it would be better to briefly introduce the most popular approaches for detection of RNA modifications (RMs), such as the m6A-seq/MeRIP-seq for transcriptome-wide profiling of m6A methylation site, and other techniques like Bisulfite sequencing for m5C detection.
3, Following the above comment, besides the wet-lab approaches, computational efforts have also contributed a lot to the development of RNA epitranscriptome study, which provides valuable datasets for lung disease-related analysis. It would be better to cover the known databases/functional tools to make this review more informative and useful to readers. Such as the most popular databases RMBase, Modomics, m6A2target, m5C-Atlas; functional tools like m6ASNP, ConsRM;
4, To study the impact of RMs specifically on chronic lung diseases, this reviewer suggested that the authors should add a section introducing the known/useful computational methods for studying functions of RNA modification in lung-related diseases, including but not limited to m6AVar (m6A-lung disease), RMVar (multiple RMs-lung diseases), RMDisaese v2.0 (multiple RMs-lung diseases) and m6A-TSHub (m6A-lung disease).
5, The information presented in the CONCLUSION section is limited. It would be more informative to summarize 1) conclusions on the RMs-lung disease studies. 2) current limitations for previous research. 3) future progress/perspectives for further exploring the potential mechanisms of lung-related diseases via epitranscriptome disturbance, such as using the 3rd sequencing techniques (e.g., DirectRMDB).
Author Response
In this presented review concerning RNA epitranscriptome in chronic lung disease, Yan et al., systematically reviewed/introduced the background/biological functions/lung disease involvement of seven well-studied RNA modifications, namely, m6A, A-to-I, m1A, m5C, Psi, m7G, and Nm. In general, the review is well-organized with serval sub-section introducing WER (writer, eraser, reader) and mechanisms/pathways of each individual modification type. Nevertheless, besides the wet-lab approaches for unveiling disease associations, many well-known computational methods have also been developed to study RNA modification in lung disease, including lung cancer. I have the following comments and hope to further improve the review quality.
1, In the manuscript, the Figures are very clear and informative. It would be even better to include some Tables for summarizing/concluding the review finding. Tables are very informative/useful in REVIEW article.
Response: We thank the reviewer for the suggestion. For the answer to this question, we have added a new table (Table 1) for summarizing the review.
2, For INTRODUCTION section, it would be better to briefly introduce the most popular approaches for detection of RNA modifications (RMs), such as the m6A-seq/MeRIP-seq for transcriptome-wide profiling of m6A methylation site, and other techniques like Bisulfite sequencing for m5C detection.
Response: We thank the reviewer for the suggestion. For the answer to this question, we have added it in line 53-65 on page 2-3 (in red).
3, Following the above comment, besides the wet-lab approaches, computational efforts have also contributed a lot to the development of RNA epitranscriptome study, which provides valuable datasets for lung disease-related analysis. It would be better to cover the known databases/functional tools to make this review more informative and useful to readers. Such as the most popular databases RMBase, Modomics, m6A2target, m5C-Atlas; functional tools like m6ASNP, ConsRM;
Response: We thank the reviewer for the suggestion. For the answer to this question, we have added it in line177-199 on page7 (in red).
4, To study the impact of RMs specifically on chronic lung diseases, this reviewer suggested that the authors should add a section introducing the known/useful computational methods for studying functions of RNA modification in lung-related diseases, including but not limited to m6AVar (m6A-lung disease), RMVar (multiple RMs-lung diseases), RMDisaese v2.0 (multiple RMs-lung diseases) and m6A-TSHub (m6A-lung disease).
Response: We thank the reviewer for the suggestion. For the answer to this question, we have added it in line199-226 on page7-8 (in red).
5, The information presented in the CONCLUSION section is limited. It would be more informative to summarize 1) conclusions on the RMs-lung disease studies. 2) current limitations for previous research. 3) future progress/perspectives for further exploring the potential mechanisms of lung-related diseases via epitranscriptome disturbance, such as using the 3rd sequencing techniques (e.g., DirectRMDB).
Response: We thank the reviewer for the suggestion. For the answer to this question, we have added it in the final Conclusions and future perspectives (in red).
Round 2
Reviewer 1 Report
"Three proteins" is totally different with "three classes of proteins"
Reviewer 2 Report
The authors have addressed my previous concerns.
Reviewer 3 Report
The authors have successfully addressed my suggestions/comments.